# Retroperitoneal Lymph Node Dissection in Colorectal Cancer with Lymph Node Metastasis: A Systematic Review

**DOI:** 10.3390/cancers15020455

**Published:** 2023-01-10

**Authors:** Michael G. Fadel, Mosab Ahmed, Gianluca Pellino, Shahnawaz Rasheed, Paris Tekkis, David Nicol, Christos Kontovounisios, Erik Mayer

**Affiliations:** 1Department of Surgery and Cancer, Imperial College London, London SW7 2AZ, UK; 2Department of Colorectal Surgery, Chelsea and Westminster Hospital NHS Foundation Trust, London SW10 9NH, UK; 3Department of Advanced Medical and Surgical Sciences, University of Campania “Luigi Vanvitelli”, 80138 Naples, Italy; 4Colorectal Unit, Vall d’Hebron University Hospital, 08035 Barcelona, Spain; 5Department of Colorectal Surgery, Royal Marsden NHS Foundation Trust, London SW3 6JJ, UK; 6Department of Academic Urology, Royal Marsden NHS Foundation Trust, London SW3 6JJ, UK

**Keywords:** lymph node dissection, colorectal cancer, metastasis, disease-free survival, recurrence

## Abstract

**Simple Summary:**

Retroperitoneal lymph node metastasis (RPLNM) occurs in up to 6% of colorectal cancer (CRC) patients. In general, there is no consensus on the treatment paradigm or optimal management of retroperitoneal lymph node dissection (RPLND) in CRC patients, necessitating a systematic review of the literature to evaluate preoperative imaging modalities, perioperative chemotherapy and radiotherapy regimens, and oncological outcomes of RPLND in CRC. Nineteen studies of 541 patients were included. Based on this systematic review and analysis, RPLND is a feasible treatment option with limited morbidity and possible oncological benefit for both synchronous and metachronous RPLNM in CRC. Future prospective clinical trials are required in order to establish further evidence for RPLND in the context of RPLNM in CRC.

**Abstract:**

The benefits and prognosis of RPLND in CRC have not yet been fully established. This systematic review aimed to evaluate the outcomes for CRC patients with RPLNM undergoing RPLND. A literature search of MEDLINE, EMBASE, EMCare, and CINAHL identified studies from between January 1990 and June 2022 that reported data on clinical outcomes for patients who underwent RPLND for RPLNM in CRC. The following primary outcome measures were derived: postoperative morbidity, disease free-survival (DFS), overall survival (OS), and re-recurrence. Nineteen studies with a total of 541 patients were included. Three hundred and sixty-three patients (67.1%) had synchronous RPLNM and 178 patients (32.9%) had metachronous RPLNM. Perioperative chemotherapy was administered in 496 (91.7%) patients. The median DFS was 8.6–38.0 months and 5-year DFS was 24.4% (10.0–60.5%). The median OS was 25.0–83.0 months and 5-year OS was 47.0% (15.0–87.5%). RPLND is a feasible treatment option with limited morbidity and possible oncological benefit for both synchronous and metachronous RPLNM in CRC. Further prospective clinical trials are required to establish a better evidence base for RPLND in the context of RPLNM in CRC and to understand the timing of RPLND in a multimodality pathway in order to optimise treatment outcomes for this group of patients.

## 1. Introduction

Colorectal cancer (CRC) is the second most common cause of cancer-related deaths worldwide [1,2]. Approximately 20% of CRC patients present with metastases at initial diagnosis, and nearly 50% of patients develop metastases, contributing to the high mortality rate [3]. In metastatic CRC, hepatic and pulmonary metastectomy has resulted in long-term survival benefit, with five-year survival rates of 25–50% and 40–68% respectively [4,5,6,7,8]. However, the precise management of retroperitoneal lymph node metastasis (RPLNM) in CRC, which occurs in up to 6% of patients [1], remains unclear.

Retroperitoneal recurrence can be accompanied by metastasis to other sites, and is associated with a poor prognosis, with reported survival rates of 31% at one year, 7.9% at two years, and 0.9% at four years [5,9]. Retroperitoneal lymph nodes in CRC are traditionally considered as non-regional nodes or distant metastasis, as classified by the American Joint Committee on Cancer [10]. It has been argued, however, that retroperitoneal nodes are in fact a continuation of the mesenteric nodal lymphatic drainage, and therefore should be considered for curative resection [11,12]. Due to this uncertain classification of retroperitoneal recurrence, views on the benefits of surgical intervention in the form of retroperitoneal lymph node dissection (RPLND) remain ambiguous.

RPLND is established in urological germ cell tumours as standard of care, and there is emerging evidence in gynaecological, gastric, and pancreatic malignancies [13,14,15]. However, the surgical morbidity and oncological benefits of RPLND in CRC have not yet been fully established, and this extensive surgical approach remains controversial. Potential complications include ileus, chylous ascites, major haemorrhage or vascular injury, acute respiratory distress syndrome, neuropraxia, and mortality [15].

Nevertheless, there has been a growing trend in performing RPLND in patients presenting with RPLNM in CRC. Early research has shown that surgery for retroperitoneal recurrence can have survival benefits compared to non-surgical management and has acceptable postoperative morbidity [16,17]. RPLND can generally be performed either above (type A) or below (type B) the renal vessels, or both [18]. Synchronous RPLND involves RPLNM identified and resected concurrently with the primary CRC, while metachronous RPLND refers to RPLNM identified and resected after surgery for primary CRC [19].

In general, there is no consensus on the treatment paradigm or optimal management of RPLND in CRC patients, necessitating a systematic review of the literature to evaluate oncological outcomes of RPLND in CRC. The main outcome measures assessed were postoperative morbidity and mortality, disease free-survival (DFS), overall survival (OS), and re-recurrence (local or distant to the RPLND surgical field). Preoperative imaging modalities, perioperative chemotherapy, and radiotherapy (RT) regimens from included studies are also presented.

## 2. Materials and Methods

### 2.1. Search Strategy

A systematic review of the literature involving RPLND in CRC was conducted according to the protocol previously published by the Cochrane collaboration. The MEDLINE, EMBASE, EMCare, and CINAHL databases were searched for studies published between January 1990 and June 2022 in the English language. The search was performed on 10th June 2022. The following medical subject headings (MeSH) and keywords were used: ‘colorectal cancer’, ‘colon’, ‘rectum’, ‘adenocarcinoma’, ‘retroperitoneal’, ‘para-aortic’, ‘synchronous’, ‘metachronous’, ‘lymph node metastasis’, ‘lymph node dissection’, ‘lymphadenectomy’, and ‘recurrence’. The search strategy for each database is shown in Appendix A. A manual search of the references from selected articles was performed to identify further relevant studies.

The systematic review was conducted according to the Preferred Reporting Items for Systematic Reviews and Meta-Analysis (PRISMA) guidelines [20] and the Cochrane Handbook for Systematic Reviews of Interventions [21]. The work was registered in the PROSPERO database for systematic reviews in December 2021 (CRD42021294057).

### 2.2. Study Selection and Inclusion Criteria

Study types included were randomised controlled trials (RCTs), retrospective or prospective cohort studies, and case-control studies. The studies chosen had to specifically relate to: (1) RPLND in patients presenting with synchronous or metachronous RPLNM in CRC; (2) RPLND for isolated RPLNM or combined with other CRC metastatic lesions; (3) pathologically positive or negative RPLNM; or (4) reporting of either postoperative morbidity or oncological outcomes.

Studies were excluded if they were: (1) articles published in a non-English language or in a book; (2) letters to the editor, case reports, or conference abstracts; (3) lacking relevant morbidity or oncological outcomes.

### 2.3. Data Extraction

The titles and abstracts were assessed by two independent authors (M.G.F. and M.A.) against the inclusion and exclusion criteria in order to arrive at a final list of articles. Any disagreement was resolved by a third independent reviewer (C.K.). Each included manuscript was read to determine ultimate inclusion in the final analysis. From the manuscripts, the following information was extracted: author names, title, year of publication, country, study centre, study design, age, gender, primary tumour location, tumour differentiation, TNM staging and R0 resection for primary tumour, and chemotherapy and RT regimen. Results were stratified by three main groups: (1) synchronous RPLNM; (2) metachronous RPLNM; and (3) synchronous and metachronous RPLNM (presented as combined outcome data in studies). Preoperative imaging modality (computed tomography, CT; magnetic resonance imaging, MRI; positron emission tomography, PET; lymph node biopsy) was recorded to define the extent of RPLNM, information on patient selection criteria, anatomical boundaries of RPLND, number of lymph nodes harvested, and lymph node yield. Outcome measures were hospital length of stay, postoperative morbidity (Clavien-Dindo classification [22,23]), postoperative 90-day mortality, DFS, OS, re-recurrence rate and site. The median, 3-year, and 5-year DFS and OS were recorded where available.

### 2.4. Quality Assessment of Studies

The quality of all observational studies was assessed using the Newcastle–Ottawa Scale [24]. This was calculated by examining three factors: method of patient selection, comparability of the study groups, and number of outcomes reported. The full score was nine stars, and studies that had a score of seven stars or more were considered high quality. All studies were rated independently by two authors (M.G.F. and M.A.), with any differences resolved by consensus.

### 2.5. Statistical Analysis

The studies were assessed for information regarding the number of patients that underwent RPLND and their associated morbidity and mortality. The data were recorded in summary tables and divided into three groups: (1) synchronous RPLNM; (2) metachronous RPLNM; and (3) synchronous and metachronous RPLNM. The mean, median, range, and standard deviation were reported where applicable. The median value of the 3-year DFS and OS and the 5-year DFS and OS were calculated. 

## 3. Results

The initial database search and additional records identified 1085 publications. A total of 1012 publications were excluded after title and abstract review and removal of duplicates. Seventy-three articles were fully reviewed, and 19 studies [9,16,17,25,26,27,28,29,30,31,32,33,34,35,36,37,38,39,40] met the criteria and were included in the final analysis. The PRISMA diagram of the literature search is demonstrated in Figure 1.

One RCT [25] and 18 retrospective cohort studies were included. Eight studies [25,26,27,28,29,30,31,32] presented results on synchronous RPLNM (290 patients), six studies [16,33,34,35,36,37] on metachronous RPLNM (129 patients), and five studies [9,17,38,39,40] on RPLND for both synchronous and metachronous RPLNM (122 patients). All studies were single-centre, and all non-randomised studies were considered to be high quality based on the Newcastle–Ottawa Scale (Appendix A).

### 3.1. Patient Demographics and Primary CRC Histopathology

Across the 19 studies, a total of 541 patients were included, with the mean age ranging from 50.0 to 68.8 years (Table 1) and 49.0% being female. There was a higher proportion of primary tumours located in the colon compared to the rectum, with 392 patients (76.4%) versus 149 patients (23.6%), respectively. Data on the TNM stage of the primary tumour were sporadically reported in 11 studies [9,16,25,26,28,29,30,31,34,36,37]. Nonetheless, where available, it was noted that at least 255 patients had advanced (T3/T4) primary tumours. Twelve studies [9,25,26,27,28,29,30,31,32,34,37,40] reported on histopathological differentiation of the primary CRC; well and moderately differentiated cancers were the most common (78.3%). Eight studies [16,25,27,28,33,34,35,37] reported on the proportion of patients achieving R0 resections for their primary CRC, with all reporting that 100% of patients achieved microscopically clear resection margins.

### 3.2. Chemotherapy and Radiotherapy Regimens

Perioperative chemotherapy was administered in all studies except for one study by Bowne et al. [33] (Table 2). Neoadjuvant chemotherapy was used less frequently compared with adjuvant chemotherapy, with 128 patients (23.7%) versus 368 patients (68.0%). The most widely used chemotherapeutic agent was 5-fluorouracil (5-FU), with several other studies reporting the application of biological agents such as bevacizumab and cetuximab. RT was used less commonly when compared to chemotherapy, with nine studies [16,17,26,30,31,35,36,37,39] reporting either neoadjuvant (34 patients, 6.3%) or adjuvant (29 patients, 5.4%) RT. RT administration was declared in a smaller proportion of studies on patients with synchronous RPLNM in comparison to the other two groups. The precise RT regimen was only reported in three studies [20,35,37], where the cumulative RT doses ranged from 45 to 55.4 Gy.

### 3.3. RPLNM Diagnostic Imaging Criteria and Selection for Surgery

CT imaging was used in all studies, either alone or in combination with an additional diagnostic modality (Table 3). Six studies [26,28,30,32,37,38] detailed their radiological criteria for CT imaging, with the general paradigm being an increased short-axis diameter and/or irregular margins signifying pathological lymph nodes. Thirteen studies [9,26,27,28,29,31,32,33,34,37,38,39,40] confirmed the use of PET-CT imaging. The following diagnostic criteria for PET-CT was used: positive/hot uptake of the radiotracer, a maximum standardised uptake value ≥5, and a lymph node diameter ≥ 10 mm or with an irregular shape on combination PET-CT. Only four studies [9,34,36,37] reported the routine use of MRI to detect RPLNM. Lymph node biopsy was performed in three studies [9,34,36]; however, all three studies reported using lymph node biopsy only in a subset of patients for whom the results of radiological investigations were not sufficient to diagnose RPLNM.

Fifteen studies [17,25,26,27,28,29,30,31,32,35,36,37,38,39,40] reported the patient selection criteria for RPLND; eight studies stated that RPLNM must be located below the renal veins (type B) to be selected for surgery [26,27,28,29,30,31,32,40], and five studies [27,31,35,37,39] specifically stated that the decision to proceed with surgery was agreed on in multidisciplinary team (MDT) meetings. Two studies [17,35] reported the need for patients to be responsive to chemotherapy and/or RT prior to surgery (Appendix A).

### 3.4. RPLND Timing and Harvesting

Synchronous and metachronous RPLND were performed in 363 patients (67.1%) and 178 patients (32.9%), respectively. The median disease-free interval for studies reporting on metachronous RPLND ranged from 12.0 to 24.4 months. Thirteen studies [9,26,27,28,29,30,31,32,34,37,38,39,40] detailed the boundaries of RPLND, with only two studies [34,39], with a total of six patients (1.1%), performing RPLND above the left renal vein (type A) (Table 3). The median and mean number of lymph nodes harvested were 12–36 and 6.9–35.6, respectively. Lymph node ratio (ratio of positive lymph nodes to total harvested lymph nodes) could be derived for eight studies [9,26,28,29,30,31,36,39], and ranged from 15.9% to 56.5%.

### 3.5. Safety and Long-Term Oncological Outcomes

Median hospital length of stay ranged from 11.0 to 40.0 days, and postoperative morbidity ranged from 8.0% to 52.1%. In total, 16 complications (12.5%) were Clavien-Dindo grade 3 and two complications (1.6%) were Clavien-Dindo grade 4 (Appendix A). Postoperative 90-day mortality was reported by 14 studies [9,16,17,25,26,27,29,30,31,32,34,36,38,39,40], with only one death recorded (Table 4).

The median follow-up duration amongst the 19 studies was 24.2 to 85.0 months. The median DFS ranged from 8.6 to 38.0 months (Table 5). The median 3-year and 5-year DFS were 21.6% (ranging from 8.9% to 49.0%) and 24.4% (ranging from 10.0% to 60.5%), respectively. The median OS was 25.0 to 83.0 months. The median 3-year OS and 5-year OS were 62.3% (ranging from 39.0 to 81.0%) and 47.0% (ranging from 15.0% to 87.5%), respectively. The overall rate re-recurrence was 27.4% to 100%. With regard to the re-recurrence site, liver, lungs, and within the local RPLND surgical field were found to be the primarily involved sites (Appendix A).

## 4. Discussion

The optimal treatment strategy for RPLNM in CRC is not yet fully established due to the low incidence of isolated RPLNM, differing views on the classification of RPLNM, and the limited evidence on associated survival outcomes and morbidity of surgery. We have presented the largest systematic review of 19 studies, comprising a total of 541 patients, in order to suggest practical recommendations for the multimodality treatment of RPLNM in CRC.

Across the 19 studies, preoperative CT imaging was routinely used followed by PET-CT imaging. Short-axis diameter > 5 mm combined with an irregular shape was the most common radiological CT criteria for RPLNM with positive fluorodeoxyglucose uptake on PET. The main patient selection factors observed included patient’s fitness for surgery, MDT discussion, clinically suspected RPLNM below the renal veins amenable to resection, and responsiveness to previous chemotherapy/RT. Perioperative chemotherapy was the preferred treatment modality, with RT only sparingly applied. Synchronous RPLND was more commonly undertaken, with the vast majority performing dissection below the renal veins. Overall, there was a reported postoperative morbidity rate of 23.3%, with a 90-day mortality rate of only 0.02% for RPLND.

It has previously been shown that the addition of PET-CT to CT imaging can improve the negative predictive value for RPLNM in cases without suspicious features on CT and the positive predictive value in cases with suspicious features on CT [41]. In light of the significant morbidity associated with RPLND, ascertaining the lymph node status of patients with suspected RPLNMs to the greatest accuracy prior to surgical decision-making is important. MRI has been found to be non-inferior to CT in detecting RPLNMs in multiple prospective studies in testicular germ cell tumours [42,43]; however, further studies are required in order to elaborate whether MRI has a defined role in RPLNMs in CRC. Regarding neoadjuvant/adjuvant therapy, it has been demonstrated that definitive chemotherapy in RPLNM can be an effective salvage treatment [44]. It must be noted here that Lee et al. [45] have concluded, following treatment of 52 patients with isolated RPLNM, that both upfront RT and deferred RT are potentially effective treatment strategies.

There have been four prior systematic reviews assessing the management (surgery ± non-surgery) of RPLNM in CRC. Ho et al. [46] included 110 patients who underwent RPLND, with a median DFS of 17–21 months and median OS of 34–44 months. At the time, they were only able to identify two case series [16,33] that included more than five patients who had surgery. Wong et al. [19] studied a 370-patient population consisting of a surgical and non-surgical group with median OS of 34–40 months and 3–14 months, respectively. Sasaki et al. [47] included 227 patients, both surgical and non-surgical groups, with the 3-year OS ranging from 60% to 100% and a median OS of 34–80 months for patients who had RPLND. The median OS was 14–42 months in the non-surgical group. Similar to the other two reviews, case reports and case series (five patients or less) were included in the analysis. Zizzo et al. [1] studied a patient population of 161 patients from nine studies with pathologically confirmed CRC isolated RPLNM who underwent RPLND. They identified a 5-year DFS rate of 0% to 60.5%, a 5-year OS rate of 53.4% to 87.5%, and a re-recurrence rate of 43.8% to 100%. In our present review of 19 studies, 541 patients underwent RPLND for isolated RPLNM (pathologically positive plus negative) with or without combined CRC metastatic lesions. A 5-year DFS rate of 10.0% to 65.0% was identified, along with a 5-year OS rate of 15.0% to 87.5% and a re-recurrence rate of 27.4% to 100%.

Regarding the surgical approach of RPLND, the dissection is often performed in the area with the following boundaries: renal hilum, bifurcation of the iliac artery, ureters bilaterally, and iliopsoas muscle [29]. This technique is challenging, particularly at the superior border, which can be a potential source of in-field recurrence. Care must be taken to identify and preserve the ureters as well as the sympathetic chain. The anatomical relationships of the lumbar vessels and sympathetic nerves in the infrarenal retroperitoneum are essential in performing a successful nerve-sparing bilateral RPLND to reduce postoperative co-morbidities and preserve nerve function [48]. Open RPLND has long been considered the standard of care. Laparoscopic RPLND, performed in three of the included studies [26,30,31], can be a technically challenging procedure requiring significant experience in laparoscopic dissection to safely access the lymph nodes posterior to the great vessels. As expertise and technology grows, in selected patients the robotic approach may provide improved ability to dissect behind the great vessels and control major bleeding more easily [49].

An emerging intraoperative technique in detecting lymph node metastases in CRC is near-infrared fluorescence (NIRF) imaging with the intention of avoiding a full (bilateral template) RPLND and its associated co-morbidities [50]. Indocyanine green (ICG) can provide visual assessment of blood vessels, blood flow, and lymph node road mapping [51,52]. Park et al. [53] used the NIRF technique for D3 lymphadenectomy in right-sided colon cancer; ICG was injected around the tumour for visualisation of lymphatic channels and lymph nodes. The number of atypical and harvested lymph nodes was significantly higher in the NIRF group compared with the conventional group. Various studies have reported on fluorescence imaging with ICG for the detection of occult CRC liver metastases with sensitivity exceeding 94% [54,55,56,57]. Furthermore, when fluorescence imaging was added to conventional imaging, extra metastases were found and resected in 20 out of 148 patients (13.5%) [58]. The potential benefits of ICG have been confirmed in other related surgical procedures, for example, in robotic-assisted radical prostatectomy with extended pelvic lymph node dissection [59]. Intraoperative confirmation and localisation of lymph nodes using fluorescents may have a role to play in increasing the accuracy of positive lymph node excision [60], and could potentially improve functional and oncological outcomes.

The presence of extracapsular invasion of resected lymph nodes for CRC has been proven to be associated with increased and earlier recurrence after CRC resection [61]. The currently included studies evaluated RPLNs on quantitative and anatomical aspects (i.e., number of positive RPLNs and their location with respect to other retroperitoneal structures). Thus, there is a need for studies to present other histological indicators, such as extracapsular invasion or extension, in order to allow for evaluation of the prognostic significance and better inform those patients who would benefit from earlier adjuvant therapy. A current phase II RCT (NCT03725254) is comparing radical surgery and retroperitoneal lymphadenectomy to radical chemoradiotherapy only for retroperitoneal lymph node recurrence of CRC, and is expected to be completed by October 2024.

### 4.1. Strengths and Limitations

This systematic review and analysis of the multimodality treatment and associated outcomes of RPLNM in CRC included a relatively large sample size of patients undergoing synchronous and metachronous RPLND to support practical recommendations on optimal surgical management.

The major limitation of this systematic review is the heterogeneity of the studies, which renders it more difficult to make firm evidence-based conclusions. The patients included were heterogenous (including isolated RPLNM with or without combined CRC metastases); certain studies reported a systematic approach to RPLND in the form of removing an entire lymph node basin, while others indicated that only grossly involved nodes were surgically removed. There were variability and differences in the patient selection process and perioperative modalities, with certain patients proceeding immediately to surgery. In general, most of the studies were single-centre retrospective studies lacking a control group, with only one RCT included.

In addition, information such as primary CRC histology, specific chemotherapy and RT regimens, and number of lymph nodes harvested were not reported in certain studies. The diagnostic criteria of CT imaging were only specified in six studies, and studies presenting sensitivity or specificity on diagnostic methods were not included. Survival rates in patients without surgery were not presented. Furthermore, the included articles ranged from 2001 to 2021, meaning that during this timeframe there have been changes and improvements in diagnostic methods, surgical techniques, and chemotherapy and RT regimens, with a growing emphasis on MDT management.

### 4.2. Implications for Multimodality Treatment of RPLNM in CRC

Based on the available evidence, practical recommendations for the surgical management of RPLNM in CRC can be proposed: all patients with suspected RPLNM should undergo PET-CT imaging prior to surgery; MRI imaging can be used in selected cases if felt necessary following MDT discussion, or where PET-CT imaging is not available; the indication and timing of chemotherapy and RT should be decided through MDT input in specialised centres; and perioperative chemotherapy may be considered to identify patients who are likely to benefit from synchronous or metachronous RPLND or to potentially improve DFS. However, the exact role of RT in the management of RPLNM is less clear based on the current evidence, and we have been unable to clearly identify whether synchronous or metachronous RPLND has superior oncological outcomes. RPLND above the renal vessels is generally avoided, highlighted in this systematic review by the patient selection process for surgery and by the observation that only six patients underwent type A RPLND. It is typically associated with high morbidity, and RPLNMs are often unresectable due to involvement of the coeliac axis, the root of the superior mesenteric artery, and adjacent organs such as the pancreas, stomach, duodenum, and renal hilum [34]. Furthermore, in these circumstances it is difficult to achieve adequate margins with irradiation.

## 5. Conclusions

RPLNM in CRC is typically considered to have a poor prognosis, with high rates of morbidity and recurrence. Based on the findings of this systematic review, it can be concluded that RPLND is a feasible treatment option with limited morbidity and possible oncological benefit for both synchronous and metachronous RPLNM in CRC. There is clearly a need for further prospective clinical trials in order to establish a better evidence base for RPLND in the context of RPLNM in CRC and to understand the timing of RPLND in a multimodality treatment pathway in order to optimise treatment outcomes for this group of patients.

## Figures and Tables

**Figure 1 cancers-15-00455-f001:**
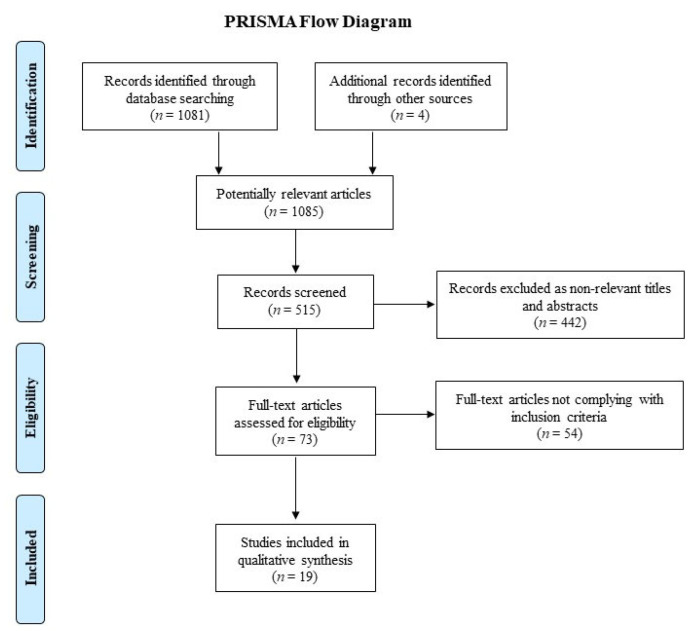
The flowchart shows the literature search and study selection process according to the PRISMA guidelines.

**Table 1 cancers-15-00455-t001:** Patient demographics, primary tumour characteristics, and quality scoring of studies [34] included in this systematic review.

Author, Year	*n*	Median/Mean Age (Range/SD)	Female, *n* (%)	Primary Tumour														Study NOS
				Location, *n* (%)		Differentiation, *n* (%)				TNM Stage, *n* (%)							R0 Resection, *n* (%)	
				Colon	Rectum	Well	Moderately	Poorly	Other	T1	T2	T3	T4	N0	N1	N2		
Synchronous RPLNM																		
Tentes et al. [25], 2007	62	―/68.8 (±10.3)	41 (66.1)	62 (100)	0	29 (46.8)	28 (45.2)	5 (8.0)	0	1 (1.6)	9 (14.5)	46 (74.2)	6 (9.7)	33 (53.2)	19 (30.6)	10 (16.1)	62 (100)	N/A *
Song et al. [26], 2016	40	―/61.7 (±10.4)	14 (35.0)	27 (67.5)	13 (32.5)	33 (82.5)		7 (17.5)		3 (7.5)		37 (92.5)		16 (40.0)	24 (60.0)		―	9
Ogura et al. [27], 2017	16	58.5/― (39–82)	11 (68.7)	14 (87.6)	2 (12.4)	11 (68.8)		5 (31.2)		―	―	―	―	―	―	―	16 (100)	9
Bae et al. [28], 2018	49	―/57.7 (±11.5)	20 (40.8)	49 (100)	0	4 (8.2)	34 (69.4)	6 (12.2)	5 (10.2)	0	1 (2.0)	43 (87.8)	5 (10.2)	―	―	―	49 (100)	7
Yamada et al. [29], 2019	36	57 (46.3–65.8)/―	15 (41.7)	17 (47.2)	19 (52.8)	8 (22.2)	19 (52.8)	2 (5.6)	7 (19.4)	0	0	11 (30.6)	25 (69.4)	―	―	―	―	8
Yamamoto et al. [30], 2019	11	―/63 (28–76)	6 (54.5)	8 (72.7)	3 (27.3)	2 (18.2)	8 (72.7)	1 (9.1)	0		1 (9.1)	8 (72.7)	2 (18.2)	5 (45.5)	1 (9.1)	5 (45.5)	―	7
Sakamoto et al. [31], 2020	29	60 (35–74)/―	14 (48.3)	14 (48.3)	15 (51.7)	2 (6.9)	19 (65.5)	6 (20.7)	2 (6.9)	0	0	13 (44.8)	16 (55.2)	0	―	―	―	9
Lee et al. [32], 2021	47	―/57.6	14 (29.8)	35 (74.5)	12 (25.5)	27 (57.4)		20 (42.6)		―	―	―	―	―	―	―	―	9
Metachronous RPLNM																		
Shibata et al. [16], 2002	20	55/―	9 (45.0)	16 (80.0)	4 (20.0)	―	―	―	―	―	1 ^†^ (4.0)	20 ^†^ (80.0)	1 ^†^ (4.0)	11 ^†^ (44.0)	10 ^†^ (40.0)	2 ^†^ (8.0)	20 (100)	8
Bowne et al. [33], 2005	16	―/―	―	16 (100)	0	―	―	―	―	―	―	―	―	―	―	―	16 (100)	8
Min et al. [34], 2008	6	―/58.2	3 (50.0)	3 (50.0)	3 (50.0)	6 (100)		0	0	0	6 (100)		0	0	6 (100)		6 (100)	9
Dumont et al. [35], 2012	23	―/51 (±8)	10 (44.0)	20 (87.0)	3 (13.0)	―	―	―	―	―	―	―	―	―	―	―	23 (100)	8
Razik et al. [36], 2014	48	60 (36–80)/―	26 (54.0)	43 (90.0)	5 (10.0)	―	―	―	―	―	―	―	―	―	23 (48.0)		―	8
Kim et al. [37], 2020	16	55.5 (42–73)/―	4 (25.0)	9 (56.3)	7 (43.8)	0	15 (93.8)	1 (6.3)	0	―	―	―	―	6 (37.5)	6 (37.5)	4 (25.0)	16 (100)	9
Synchronous and Metachronous RPLNM																		
Elias et al. [17], 2001	31	―/50 (±11)	25 (80.6)	26 (83.9)	5 (16.1)	―	―	―	―	―	―	―	―	―	―	―	―	7
Choi et al. [9], 2010	24	―/52 (27–78)	11 (45.8)	15 (62.5)	9 (37.5)	17 (70.8)		7 (29.2)		1 (4.2)	1 (4.2)	20 (83.3)	2 (8.3)	1 (4.2)	5 (20.8)	18 (75.0)	―	9
Arimoto et al. [38], 2015	14	66 (42–75)/―	3 (21.4)	6 (42.9)	8 (57.1)	―	―	―	―	―	―	―	―	―	―	―	―	8
Gagniere et al. [39], 2015	25	55 (31–69)/―	16 (64.0)	12 (48.0)	13 (52.0)	―	―	―	―	0	―	―	―	―	―	―	―	9
Ichikawa et al. [40], 2021	28	61 (42–79)/―	15 (53.6)	―	―	23 (82.1)		5 (17.9)		―	―	―	―	―	―	―	―	8

CRC, colorectal cancer; NOS, Newcastle–Ottawa Scale; RPLNM, retroperitoneal lymph node metastasis; SD, standard deviation; † out of a total of 25 patients, 5 of whom were not included in outcome analysis by authors; ―, not reported; * Newcastle–Ottawa Scale not applicable as randomised controlled trial.

**Table 2 cancers-15-00455-t002:** Chemotherapy and radiotherapy details for pre- and post-retroperitoneal lymph node dissection in colorectal cancer.

Author, Year	Chemotherapy, *n* (%)		Chemotherapy Regimens	Radiotherapy, *n* (%)		Radiotherapy Regimens
	Pre-RPLND	Post-RPLND		Pre-RPLND	Post-RPLND	
Synchronous RPLNM						
Tentes et al. [25], 2007	―	30 (48.4)	5-FU (500 mg/m^2^) combined either with leucovorin (200 mg/m^2^) or isovorin (175 mg/m^2^)	―	―	―
Song et al. [26], 2016	7 (17.5)	24 (60.0)	Adjuvant regimens = (i) 5-FU; (ii) capecitabine based ± oxaliplatin or irinotecan	7 (17.5)	―	―
Ogura et al. [27], 2017	4 (25.0)	15 (93.8)	Adjuvant regimens (i) oxaliplatin or irinotecan, *n* = 10 (62.5); (ii) Other, *n* = 6 (37.5)	―	―	―
Bae et al. [28], 2018	0	47 (95.9)	Every 3–4 weeks for 6 months: (i) 5-FU + leucovorin (ii) FOLFOX	―	―	―
Yamada et al. [29], 2019	2 (5.6)	25 (69.4)	―	―	―	―
Yamamoto et al. [30], 2019	1 (9.1)	6 (54.5)	―	1 (9.1)	―	―
Sakamoto et al. [31], 2020	1 (3.4)	17 (58.6)	―	0	2 (6.9)	―
Lee et al. [32], 2021	0	38 (80.9)	Adjuvant regimens (i) 5-FU, *n* = 2 (4.3); (ii) 5-FU + oxaliplatin, *n* = 6 (12.8); (iii) 5-FU + irinotecan, *n* = 4 (8.5); (iv) 5-FU + oxaliplatin + irinotecan, *n* = 25 (53.2)Biological agents (i) Bevacizumab, *n* = 7 (14.9); (ii) Cetuximab, *n* = 3 (6.4); (iii) Bevacizumab + cetuximab, *n* = 15 (31.9)	―	―	―
Metachronous RPLNM						
Shibata et al. [16], 2002	6 (30.0)	14 (70.0)	―	4 (20.0)	5 (25.0)	―
Bowne et al. [33], 2005	―	―	―	―	―	―
Min et al. [34], 2008	―	6 (100)	5-FU, leucovorin and oxaliplatin	―	―	―
Dumont et al. [35], 2012	19 (83.0)	23 (100)	(i) LV5FU2, *n* = 12 (52); (ii) LV5FU2 plus oxaliplatin, irinotecan, cetuximab and/or bevacizumab, *n* = 11 (48.0)	4 (17)	5 (22.0)	45–50 Gy in ‘normofractionated’ ^#^
Razik et al. [36], 2014	20 (41.7)	8 (16.7)	―	18 (37.5)	0	―
Kim et al. [37], 2020	0	13 (81.3)	―	0	4 (25.0)	48–55.4 Gy in 25–31 ^#^
Synchronous and Metachronous RPLNM						
Elias et al. [17], 2001	31^a^ (100)		5-FU + folinic acid over 6 month period	0	12 (38.7)	45 Gy
Choi et al. [9], 2010	―	23 (95.8)	(i) 5-FU + leucovorin based or capecitabine based, *n* = 13(54.2); (ii) Oxaliplatin or irinotecan based, *n* = 10 (41.7)	―	―	―
Arimoto et al. [38], 2015	9 (64.0)	4 (29.0)	Adjuvant regimens (i) FOLFOX, *n* = 3 (21.4); (ii) Capecitabine, *n* = 1 (7.1); (iii) CAPOX, *n* = 1 (7.1); (iv) uracil-tegafur + leucovorin, *n* = 1 (7.1)Neoadjuvant regimens (i) FOLFOX + bevacizumab, *n* = 5 (35.7); (ii) FOLFOX + panitumumab, *n* = 1 (7.1); (iii) CAPOX + bevacizumab, *n* = 3 (21.4)	―	―	―
Gagniere et al. [39], 2015	15 (60.0)	21 (84.0)	(i) LV5FU2; (ii) folinic acid plus oxaliplatin or irinotecan ± cetuximab or bevacizumab	0	1 (4.0)	―
Ichikawa et al. [40], 2021	13 (46.4)	23 (82.1)	(i) 5-FU + levofolinate calcium, *n* = 5 (17.9); (ii) FOLFOX/FOLFIRI ± bevacizumab, *n* = 22 (78.5)	0	0	―

^a^ all patients in study received at least two lines of specified chemotherapy either before or after surgery; CAPOX, capecitabine and oxaliplatin; FOLFIRI, irintotecan, folinic acid and fluorouracil; FOLFOX, oxaliplatin, folinic acid and 5-fluorouracil; Gy, Gray(s); LV5FU2, bolus and infusional 5-fluorouracil and leucovorin; RPLND, retroperitoneal lymph node dissection; RPLNM, retroperitoneal lymph node metastasis; 5-FU, 5-fluorouracil; ―, not reported; ^#^, fractions.

**Table 3 cancers-15-00455-t003:** Retroperitoneal lymph node metastasis diagnostic methods and radiological criteria.

Author, Year	Retroperitoneal Lymph Node Metastasis Diagnostic Methods and Radiological Criteria						
	Biopsy	CT		MRI		PET	
		Usage	Criteria	Usage	Criteria	Usage	Criteria
Synchronous RPLNM							
Tentes et al. [25], 2007	N	Y	―	N	N/A	N	N/A
Song et al. [26], 2016	N	Y	Short diameter >8 mm, irregular margin or central necrosis	N	N/A	Y	Positive FDG uptake
Ogura et al. [27], 2017	N	Y	―	N	N/A	Y	Hot FDG uptake
Bae et al. [28], 2018	N	Y	5 mm short-axis diameter, with spiculated borders or showing a mottled heterogenic pattern	N	N/A	Y	Positive FDG uptake
Yamada et al. [29], 2019	N	Y	―	N	N/A	Y	―
Yamamoto et al. [30], 2019	N	Y	Shorter diameter >8 mm, irregular margin or heterogeniccontrast pattern	N	N/A	N	N/A
Sakamoto et al. [31], 2020	N	Y	―	N	N/A	Y	―
Lee et al. [32], 2021	N	Y	Diameter ≥10 mm or irregular shape (PET-CT)	N	N/A	Y	Diameter ≥10 mm or irregular shape (PET-CT)
Metachronous RPLNM							
Shibata et al. [16], 2002	N	Y	―	N	N/A	N	N/A
Bowne et al. [33], 2005	N	Y	―	N	N/A	Y	―
Min et al. [34], 2008	Y	Y	―	Y	―	Y	Positive FDG uptake
Dumont et al. [35], 2012	N	Y	―	N	N/A	N	N/A
Razik et al. [36], 2014	Y	Y	―	Y	―	N	N/A
Kim et al. [37], 2020	N	Y	Short axis diameter >8 mm	Y	Short axis diameter >8 mm	Y	High FDG uptake
Synchronous and Metachronous RPLNM							
Elias et al. [17], 2001	N	Y	―	N	N/A	N	N/A
Choi et al. [9], 2010	Y	Y	―	Y	―	Y	―
Arimoto et al. [38], 2015	N	Y	Minor axis diameter >5 mm	N	N/A	Y	Maximum standardised uptake value ≥5.0
Gagniere et al. [39], 2015	N	Y	―	N	N/A	Y	―
Ichikawa et al. [40], 2021	N	Y	―	N	N/A	Y	High FDG uptake

CT, computed tomography scan; FDG, fluorodeoxyglucose; MRI, magnetic resonance imaging; N/A, not applicable; N, no; PET, positron emission tomography scan; RPLNM, retroperitoneal lymph node metastasis; Y, yes; ―, not reported.

**Table 4 cancers-15-00455-t004:** Location and timing of retroperitoneal lymph node dissection in colorectal cancer and information on lymph node harvesting.

Author, Year	RPLN Locations		Median DFI in Metachronous Cases, Months (Range)	Timing of RPLND		Lymph Nodes Harvested		
	Type A, *n* (%)	Type B, *n* (%)		Synchronous, *n* (%)	Metachronous, *n* (%)	Median/Mean, *n* (Range/SD)	Median/Mean Pathologically Positive RPLNs, *n* (Range/SD)	Lymph Node Ratio, %
Synchronous RPLNM								
Tentes et al. [25], 2007	―	―	N/A	62 (100)	0	―/19 ^a^ (6–61)	―/―	―
Song et al. [26], 2016	0	40 (100)	N/A	40 (100)	0	―/6.9 (1–29/±6.6)	―/1.1 (0–17/±2.8)	15.9
Ogura et al. [27], 2017	0	16 (100)	N/A	16 (100)	0	20 ^a^ (13–38)/―	1 (0–4)	―
Bae et al. [28], 2018	0	49 (100)	N/A	49 (100)	0	―/6.9 ^b^ (±5.2)	―/3.9 ^b^ (±4.0)	56.5 ^b^
Yamada et al. [29], 2019	0	36 (100)	N/A	36 (100)	0	36 (8–99)/―	13/―	35.0
Yamamoto et al. [30], 2019	0	11 (100)	N/A	11 (100)	0	―/8 ^b^ (1–23)	4 ^b^ (1–23)/―	50.0
Sakamoto et al. [31], 2020	0	29 (100)	N/A	29 (100)	0	12 ^b^ (1–81)/―	4 ^b^ (1–71)/―	33.0
Lee et al. [32], 2021	0	47 (100)	―	47 (100)	―	―/35.6 ^a^ (±19.2)	―	―
Metachronous RPLNM								
Shibata et al. [16], 2002	―	―	23 (3–72)	0	20 (100)	―	―	―
Bowne et al. [33], 2005	―	―	―	0	16 (100)	―	―	―
Min et al. [34], 2008	2 (33.3)	4 (66.7)	22 ^†^	0	6 (100)	―	―	―
Dumont et al. [35], 2012	―	―	22 ^†^	0	23 (100)	―/14 (±14)	―/7 (±11)	50.0
Razik et al. [36], 2014	―	―	22 (3–270)	0	48 (100)	―	―	―
Kim et al. [37], 2020	0	16 (100)	24.4 ^†^ (±12.5)	0	16 (100)	―	1 (1–6)	―
Synchronous and Metachronous RPLNM								
Elias et al. [17], 2001	―	―	―	10 (32.3)	21 (67.7)	―/16 (3–53/±13)	―/8.5 (1–49/±7)	53.0
Choi et al. [9], 2010	0	24 (100)	―	19 (79.2)	5 (20.8)	―	―	―
Arimoto et al. [38], 2015	0	14 (100)	―	9 (64.3)	5 (35.7)	―	―	―
Gagniere et al. [39], 2015	4 (16.0)	21 (84.0)	12 (5–42)	19 (76.0)	6 (24.0)	21 (4–56)/―	4 (1–41)/―	19.0
Ichikawa et al. [40], 2021	0	28 (100)	―	16 (57.1)	12 (42.9)	―	―	―

^a^ all harvested lymph nodes; ^b^ para-aortic lymph nodes only; DFI, disease-free interval; RPLN, retroperitoneal lymph node; RPLND, retroperitoneal lymph node dissection; RPLNM, retroperitoneal lymph node metastasis; SD, standard deviation; ^†^ mean value; ―, not reported.

**Table 5 cancers-15-00455-t005:** Postoperative morbidity and mortality, disease-free, overall survival, and re-recurrence outcomes.

Author, Year	Median/Mean LOS, Days (Range/SD)	Morbidity, *n* (%)	Mortality, *n* (%)	Median Follow-Up Duration, Months (Range)	Disease-Free Survival			Overall Survival			Re-recurrence, *n* (%)	Re-recurrence Sites	
					Median/Mean, Months (Range)	3 Year, %	5 Year, %	Median/Mean, Months (Range/SD)	3 Year, %	5 Year, %		Including RPLND Field, *n* (%)	Not Including RPLND Field, *n* (%)
Synchronous RPLNM													
Tentes et al. [25], 2007	―/―	11 (17.7)	1 (1.6)	―	―/―	―	―	―/94 (±6)	―	75.0	17 (27.4)	5 (8.1)	12 (19.4)
Song et al. [26], 2016	―/9.8 (±5.7)	6 (15.0)	0	31 (9.1–103.1)	―/―	40.2 ^b^	―	―/―	65.7 ^b^	―	9 ^b^ (56.3)	4 ^b^ (10)	5 (12.5)
Ogura et al. [27], 2017	―/―	3 (18.8)	0	58.8 (2.4–103.2)	―/―	―	60.5 (RFS)	―/―	―	70.3 (CSS)	7 (43.8)	4 (25.0)	3 (18.8)
Bae et al. [28], 2018	―/―	―	―	―	―/―	―	26.5	37 (6–169)/―	―	33.9	―	―	―
Yamada et al. [29], 2019	24.5 (14–429)/―	14 (38.9)	0	25.2 (10.8–62.4)	―/―	―	22.2 (RFS)	―/―	―	25.0	29 (80.6)	9 (26.0)	―
Yamamoto et al. [30], 2019	11 (7–19)/―	3 (27.3)	0	―	17 (2–44)/―	―	―	25 (2–44)/―	―	―	4 (36.4)	1 (9.1)	4 (36.4)
Sakamoto et al. [31], 2020	40 (8–106)/―	9 (31.0)	0	30 (1.5–210)	―/―	17.2 (RFS)	―	―/―	50.5	―	23 (79.3)	2 (6.9)	―
Lee et al. [32], 2021	―/20.8	18 (38.3)	―	27	―/―	―	―	―/―	―	33.9	34 (72.3)	6 (12.8)	―
Metachronous RPLNM													
Shibata et al. [16], 2002	―/―	5 (25.0)	0	29 (1–151)	17/―	―	10	40 (4–151)/―	―	15.0	12 (60.0)	11 (55.0)	―
Bowne et al. [33], 2005	―/―	―	―	27 ^c^	―/―	―	―	44 (23–66) ^a^/―	―	―	―	―	―
Min et al. [34], 2008	―/―	2 (33.3)	0	30 ^c^	21/28	―	―	34/―	―	―	6 (100)	0	6 (100)
Dumont et al. [35], 2012	―/―	―	―	47 (4–258)	―/―	26	―	53 (4–258)/―	81.0	―	―	―	―
Razik et al. [36], 2014	―/―	25 (52.1)	0	32	38/―	―	49	80/―	―	70.0	21 (48.8)	8 ^†^ (16.7)	14 ^†^ (29.2)
Kim et al. [37], 2020	―/―	―	―	50 (30–72) ^‡^	36 (9–144)/―	―	―	83 (32–182)/―	―	87.5	8 (50.0)	3 (18.8)	5 (31.2)
Synchronous and Metachronous RPLNM													
Elias et al. [17], 2001	―/―	6 (19.4)	0	24.2 (6–120)	―/17	9.6	―	―/―	39.0	―	26 (83.8)	6 (19.4)	20 (64.5)
Choi et al. [9], 2010	13.8 (7–30)	5 (27.8)	0	29 (7–75)	14 (DFI)/―	49	22	64 (17–111)/―	59.4	53.4	16 (66.7)	7 (29.2)	9 (37.5)
Arimoto et al. [38], 2015	―/―	7 (50.0)	0	33.2 (4.3–50.6)	8.6/―	8.9	―	36.1 (8.7–70.8)/―	62.3	―	12 (86.0)	―	―
Gagniere et al. [39], 2015	16 (17–23)	2 (8.0)	0	85 (4–152)	―/―	―	―	60 (4–142)/―	64.0	47.0	15 (60.0)	13 (52.0)	―
Ichikawa et al. [40], 2021	22.5 (12–87)/―	10 (35.7)	0	―	―/―	―	―	―/―	―	21.4	23 (82.1)	11 (39.3)	―

^a^ 95% confidence interval; ^b^ only includes patients with positive para-aortic lymph nodes on biopsy (*n* = 16); ^c^ whole cohort including those who did not have RPLND; CSS, cancer-specific survival rate; DFI, disease-free interval; LOS, length of hospital stay; RFS, relapse/recurrence-free survival rate; RPLND, retroperitoneal lymph node dissection; RPLNM, retroperitoneal lymph node metastasis; SD, standard deviation; ^†^ authors unable to retrieve site of re-recurrence in 3 patients; ^‡^ interquartile range; ―, not reported.

## Data Availability

The datasets generated and analysed during the current study are available from the corresponding author upon request.

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
