# Peer review of "Retroperitoneal Lymph Node Dissection in Colorectal Cancer with Lymph Node Metastasis: A Systematic Review"

_cancers, 2023, doi:10.3390/cancers15020455_

Round 1

Reviewer 1 Report

Excellent work this very challenging and hot topic in colorectal topic.

I have read this manuscript and is an excellent well written piece of work that addresses the review question.  It is true that there is no consensus in the treatment paradigm or optimum management of RPLND in CRC patients necessitating a systematic review of the literature - the authors manage to do that in a well-structured and presented systematic review.

Reviewer 2 Report

The paper by Fadel et al. Is well written on very important subject. 

I do have some comments: 

The aim was to evaluate the outcomes for CRC patients undergoing RPLND - and accordingly the search method with the keywords specified on page 2, line 83-89 was used. However, much emphases has been put on evaluating preoperative imaging, chemotherapy and radiotherapy regimes without a thorough systematic review. 

The authors also make strong recommendations for surgery in the Simple Summary, that cannot be supported by the excising litterateur with mostly small cohort studies. 

Page 1, line 22, Rephrase the sentence to better match with the conclusion in abstract. Possible change in treatment favouring surgery is not supported by the review. 

Rephrase sentence on page 2, line 73-75 “a systematic review to evaluate preoperative imaging modalities, preoperative ….” A suggestion is to start with the main outcome and include that also imaging modalities etc. from included studies are presented. 

Discussion page 11, sentence 121. Please include in more detail the main differences between the present Review and Review by Zizzo et al. 

In section 4.1 please include that only in six studies CT criteria were specified and studies presenting sensitivity/specificity on diagnostic methods are not included. Survival rates in patients without surgery have not been presented. 

In section 4.2 sentence 191 “perioperative chemotherapy may be considered” based on what? In patients with meta or synchronous disease? 

Sentence 195 “RPLND above the renal vessels should be avoided due to moribidity” I do agree however based on current paper only 6 patients are included and the morbidity rates comparable with Type B RPLN. 
